# Effect of Pretreatment and Cryogenic Temperatures on Mechanical Properties and Microstructure of Al-Cu-Li Alloy

**DOI:** 10.3390/ma14174873

**Published:** 2021-08-27

**Authors:** Cheng Wang, Jin Zhang, Youping Yi, Chunnan Zhu

**Affiliations:** 1Light Alloy Research Institute, Central South University, Changsha 410083, China; peterchenglari@csu.edu.cn (C.W.); yyp@csu.edu.cn (Y.Y.); zhuchunnan@csu.edu.cn (C.Z.); 2State Key Laboratory of High Performance and Complex Manufacturing, Central South University, Changsha 410083, China

**Keywords:** Al-Cu-Li alloy, pretreatment, cryogenic temperature, mechanical property, microstructure

## Abstract

The mechanical properties of Al-Cu-Li alloys after different pretreatments were investigated through tensile testing at 25 and −196 °C, and the corresponding microstructure characteristics were obtained through optical metallography, scanning electron microscopy, electron backscatter diffraction, and transmission electron microscopy. An increasing mechanism of both strength and ductility at cryogenic temperatures was revealed. The results show that the hot deformation pretreatment before homogenization promoted the precipitation of Al3Zr particles, improved particle distribution, and inhibited local precipitation-free zones (PFZ). Both hot deformation pretreatment before homogenization and cryogenic temperature were able to improve strength and ductility. The former improved strength by promoting the precipitation of Al3Zr particles while enhancing the strengthening effect of the second-phase particles and reducing the thickness of the coarse-grained layer. Meanwhile, the increase in ductility is attributable to the decrease in thickness of the coarse-grained layer, which reduced the deformation incompatibility between the coarse and fine grains and increased the strain-hardening index. The latter improved the strength by suppressing dynamic recovery during the deformation process, increasing the dislocation density, and enhancing the work hardening effect. Additionally, the increase in ductility is attributable to the suppression of planar slip and strengthening of grain boundaries, which promoted the deformation in grain interiors and made the deformation more uniform.

## 1. Introduction

Al-Cu-Li alloys, with high specific strength and modulus, have been rapidly developed in recent years and are widely used in the aerospace industry [1,2,3]. They replace general aluminum alloys used in the manufacture of wing skins and rocket fuel tanks for launch vehicles, and this can reduce weight by 10–20% and increase stiffness by 15–30% [4]. However, the relatively low formability at room temperature, easy wrinkle and rupture during the forming process, and easy corrosion after hot forming significantly limit their further development [5,6]. The excellent mechanical properties of the alloy not only depend on the microstructure but are also closely related to temperature [7]. The microstructure can be well regulated through an appropriate heat treatment process, thereby improving its mechanical properties [8,9,10]. Metallic materials have metallurgical heritability, and metallurgical defects of as-cast ingots have negative effects on subsequent thermo-mechanical processing and heat treatment. Metallurgical defects can be effectively eliminated by ingot homogenization, such as microsegregation during casting, dissolution of the coarse secondary phase, and regulation of the formation of dispersoids [11,12,13]. Therefore, it is crucial to regulate the distribution of fine-sized Al3Zr dispersoids by optimizing the homogenization heat treatment.

In the past decade, many scholars have explored ways to regulate the precipitation of Al3Zr particles and, subsequently, the grain structure to improve the mechanical properties of the alloy. For example, Wu et al. [14] studied the effects of four different homogenization treatments on the precipitation behavior of Al3(Er,Zr) particles and their effects on recrystallization resistance in a new alloy, Al-Zn-Mg-Er-Zr; they found that compared with traditional single-stage homogenization, a finer particle size, higher number density, and volume fraction of Al3(Er,Zr) particles can be obtained in the other three homogenization treatments. A minimized width of the precipitation-free zone near the grain boundary and a significantly lower recrystallized fraction were also observed. Tsivoulas et al. [15] studied the heterogeneous segregation of Zr solutes and the distribution of Al3Zr in Al-Cu-Li alloy through ramp heating homogenization. Al3Zr strip-shaped cluster preferential orientations along <100>Al were also found, and they were proven to be independent of the metastable phase θ’(Al2Cu). The precipitation of Al3Zr particles is based on an interaction between the Zr solute and dislocation climb after the Zr atom diffusion to dislocation.

Further, it was reported that the strength and plasticity of aluminum alloy could be cooperatively enhanced at cryogenic temperatures, which makes its application attractive in low-temperature environments [16,17,18]. In recent years, researchers have aimed to clarify the mechanism of cooperative enhancements in the strength and plasticity of aluminum alloys at cryogenic temperatures. Dong Fei et al. [19] studied the flow behaviors and microstructure evolution of WQ-tempered Al-Li alloy from −196 to 25 °C at different strain rates; they found that the plasticity increased gradually with decreases in the deformation temperature, but it remained insensitive to strain rate. In comparison with room temperature, the plastic deformation of the sample at cryogenic temperatures is more homogeneous, and grain rotation weakens the fiber texture. Meanwhile, a higher work hardening rate is obtained and local necking is delayed. Liu Wei et al. [20] found that the Al-Cu-Mn alloy exhibits cooperatively enhanced ductility and strain-hardening at cryogenic temperatures. The reason for the enhancement of ductility is the reduction in the accumulation of movable dislocations along the grain boundary and increased storage capacity in the grain. The increased strength is attributed to the decrease in the relative slip distance of the activated dislocations, and the suppression of tangles and cells of dislocation collapse during the cryogenic deformation. Yuan Shijian et al. [21] studied the deep drawability of Al-Cu-Mn alloy at room and cryogenic temperatures; they found that with the local thinning being weaker after cup drawing at cryogenic temperature, uniform thickness distribution, and a large drawing height and drawing load, the deep drawability was significantly improved. After combining the observation of the microstructure, the reason for the enhancement of deep drawability was clarified.

In this study, three different pretreatments were performed on the as-cast Al-Cu-Li alloy before homogenization. The three types of homogenized samples were hot-rolled to a thickness of 6 mm. Then, uniaxial tensile tests were carried out at 25 and −196 °C. The effects of pretreatment and cryogenic temperatures on the mechanical properties and microstructure of Al-Cu-Li alloy were examined and discussed in detail.

## 2. Materials and Methods

The chemical composition of Al-Cu-Li alloy used in this study is shown in Table 1. The process diagram of the whole heat treatment and test conditions are shown in Figure 1. First, the ingot was compressed by 10% at 400 °C to reduce the potential influence from the ingot’s pores. After deformation, the ingots were divided into three groups for different pretreatments. The first group of samples was annealed for 12 h at 300 °C, the second group of samples was not treated, and the third group was compressed by 10% at 400 °C. After pretreatment, the three groups of samples were treated with the same ramp heating homogenization: beginning at room temperature, heating for 10 h to 510 °C, maintaining an isothermal plateau for 12 h, and finally water quenching. Figure 2a shows the DSC curve of as-cast Al-Cu-Li alloy. According to the temperature of melting of the eutectic phase shown in the Figure 2a, the homogenization temperature was selected as 510 °C. Figure 2b shows that only a few dislocations appeared in the as-cast Al-Cu-Li alloy. The three groups of homogenized samples were named annealed state (AS), deformed state (DS), and thermal deformed state (TDS). The three groups of homogenized samples were heated at 510 °C for 3 h, then hot-rolled into 6 mm thick plates. The three hot-rolled plates were named annealed state plate (ASP), deformed state plate (DSP), and thermal deformed state plate (TDSP), respectively.

Tensile specimens were cut along the rolling direction, and their geometric dimensions are shown in Figure 3 (strictly according to the ISO 15579: 2000 standard). The samples were divided into two groups according to their thickness. The thickness of the first group was 6 mm (the original thickness was retained, Ot), while that of the second group was 3 mm (only the central layer was retained, Cl). After solution treatment at 510 °C for 1 h and water quenching (with a transferring time of less than 5 s), the specimens were quickly transferred to a CMT5105GL test machine (Zhuhai SUST Electrical Equipment Co., Ltd., Zhuhai, China) for tensile testing at a speed of 2 mm/min in which the measured value equaled the average value of five samples. The deformation temperatures were 25 and −196 °C, respectively. The −196 °C temperature was obtained by soaking samples in liquid nitrogen with a holding time of 10 min.

To observe the surface morphology of the samples at 0.12 strain, an Olympus DSX500 (Olympus Corporation, Tokyo, Japan) was used for optical metallography (OM) and a Zeiss EVO M10 (Zeiss, Oberkochen, Germany) was used for scanning electron microscopy (SEM) (equipped with an EBSD detector), with the grain structure and Kernel average misorientation (KAM) distribution examined. OM and SEM samples were observed after grinding and polishing. Before the tensile test, a FIB 600i double-beam scanning electron microscope (FEI, Hillsboro, OR, USA) was used to draw a grid on the surface of the SEM specimen. The grid was 100 × 100 μm^2^, spacing was 10 μm, and depth was 0.8 μm. The EBSD specimen was first polished through mechanical grinding, followed by fabric polishing, then electric polishing using a solution of 10% perchloric acid and 90% ethyl alcohol at a voltage of 20 V. The microstructural features were characterized using an FEI Titan F20 G2 (FEI, Hillsboro, OR, USA), which was operating at 200 kV. The samples for transmission electron microscopy (TEM) (FEI, Hillsboro, OR, USA) imaging analysis were prepared using mechanical grinding with a thickness of 80 mm and cut to 3 mm radius disks. Then, electropolishing was performed using a Tenupol 5 machine (Struers, Copenhagen, Denmark) with a solution of 30% nitric acid and 70% methanol at −30 to −20 °C and 15–20 V.

## 3. Results

### 3.1. Mechanical Properties

Figure 4 shows the stress–strain curves of the Ot and C1 samples of three different Al-Cu-Li alloy plates at 25 and −196 °C, with the corresponding tensile properties listed in Table 2. Figure 4a shows that at 25 and −196 °C, from ASP to DSP to TDSP, the tensile strength and elongation of three different plate samples gradually increased. The hot deformation treatment before homogenization increased the strength and elongation of the alloy. At 25 and −196 °C, in comparison with the DSP and ASP samples, the ultimate tensile strength of the Ot sample of TDSP increased by 9.4%, 3.2%, 14.8%, and 10.4%, respectively, and the elongation increased by 13.5%, 5.2%, 23.9%, and 11.0%, respectively. Compared with the Ot sample, the ultimate tensile strength and elongation of the Cl sample were higher. In comparison with 25 °C, the ultimate tensile strength and elongation of three different plate samples were improved significantly at −196 °C. From 25 to −196 °C, the ultimate tensile strength of the Ot and Cl samples of the ASP increased by 23.7% and 28.5%, respectively, and the elongation increased by 105.6% and 109.3%, respectively. The ultimate tensile strength of the Ot and Cl samples of the DSP increased by 26.2% and 30.8%, respectively, and the elongation increased by 98.7% and 97.3% respectively. The ultimate tensile strength of the Ot and Cl samples of the TDSP increased by 19.0% and 32.8%, respectively, and the elongation increased by 84.1% and 82.2%, respectively.

At 25 °C, the tensile curves of three different plate samples all showed the Portevin–Le Chatelier (PLC) effect, but this phenomenon disappeared at −196 °C, as shown in the enlarged black dashed frame in Figure 4. It is closely related to Dynamic Strain Aging (i.e., the dynamic interaction between mobile dislocations and solute atoms [22]). The mobile dislocations are pinned by the Cottrell atmospheres formed by the solute atoms, which increases the flow stress; when the applied stress exceeds the pinning force, the dislocations are unpinned, which reduces the flow stress. At −196 °C, the diffusion rate of solute atoms in the alloy is reduced, and it is difficult to form the Cottrell atmospheres, thereby weakening the PLC effect [19]. Figure 5 shows the comparison of the n value and yield strength ratios of the Ot and Cl samples for three types of plates at 25 and −196 °C. As shown in Figure 5a, the variation law of the n value of the three different plates was the same as the tensile properties, and the rank was TDSP sample > DSP sample > ASP sample; from 25 to −196 °C, the n value was significantly increased. The variation law of the yield strength ratio of the three types of plates was opposite to the n value, and the rank was ASP sample > DSP sample > TDSP sample; the yield strength ratio at −196 °C was lower than that at 25 °C, as shown in Figure 5b. The increase in the n value and the decrease in yield strength ratio can increase the formability of the plate [23].

### 3.2. Grain Structure

Figure 6 shows the optical metallographic images of three Al-Cu-Li hot-rolled plates after solution treatment at 510 °C for 1 h. Figure 6 shows that the centers of three different plates were the stripe-shaped grains, and a certain thickness of coarse-grained layer appeared on the surface layer; however, the thickness of the coarse-grained layer was different. The thickness of the coarse-grained layer was 1377 μm for the ASP sample, 574 μm for the DSP sample, and 49 μm for the TDSP sample. Generally, the heat that was transferred from the surface layer of the plate to the rollers and the air during the rolling process results in the surface layer temperature being lower than the central layer one. Meanwhile, the deformation of the surface layer was greater than that of the central layer, which leads to higher deformation stored energy of the surface grain structure, and it was easy to form a coarse-grained layer after solution treatment.

### 3.3. Fracture Morphology

Figure 7 shows the tensile fracture morphology of three different plate samples near the surface. The fracture is divided into coarse-grained zones and fine-grained zones, and secondary cracks between the coarse and fine grains are observed in the ASP and DSP samples. During plastic deformation, due to the existence of the coarse-grained layer in three different plates, the incompatible deformation between the coarse and fine grains is prone to micro-cracks; this leads to premature cracking and reduced ductility. Figure 7a–c shows that the thickness of the coarse-grained layer gradually decreased. This means that the ductility of the alloy should have increased, which is consistent with the variation of the elongation, as shown in Figure 4.

## 4. Discussion

### 4.1. Effect of Pretreatment and Cryogenic Temperature on Strength

The strengthening mechanism of Al-Cu-Li alloy includes solid solution strengthening, fine grain strengthening, second-phase particle strengthening, and work hardening. At 25 and −196 °C, the ultimate tensile strengths of the three different plate samples are significantly different. Since the composition, homogenization system, hot rolling process, and solid solution system of the three different plate samples are the same, the difference in strength mainly comes from the difference in grain size and the precipitation of the second-phase particle. Figure 8 shows the STEM images of the three different homogenized samples. More than 30 STEM micrographs are statistically analyzed with different sight fields in each specimen. The statistical results include the number density and size distribution of the Al3Zr particles, as shown in Figure 9. In the AS sample, the distribution of spherical Al3Zr particles was heterogeneous; there were more local precipitation-free zones (PFZ), as shown by the red enclosed dashed line in Figure 8a, and the number density of particles was the lowest, as shown in Figure 9d. In comparison with the AS sample, the number density of Al3Zr particles in the DS sample increased (shown in Figure 9d), while the local PFZ decreased, as shown in Figure 8b. In comparison with the DS sample, the number density of Al3Zr particles in the TDS sample further increased, and the particles presented a uniformly dense distribution in the entire field of view, as shown in Figure 8c. Meanwhile, the local PFZ disappeared. The average radii of Al3Zr particles in the three different plate samples were 15.5, 14.9, and 11.6 nm, and the particle radius gradually decreased, as shown in Figure 9a–c. Therefore, among the three different plate samples, the thickness of the coarse-grained layer of the Ot sample of TDSP was the thinnest, and the number density of Al3Zr particles was the highest; thus, the ultimate tensile strength was the highest. In comparison with the TDSP sample, the thickness of the coarse-grained layer of the Ot sample of the DSP increased, and the number density of Al3Zr particles decreased; therefore, the ultimate tensile strength was reduced. In comparison with the DSP sample, the Ot sample of the ASP had the largest thickness of coarse-grained layer, and the Al3Zr particle number density was the lowest; therefore, the ultimate tensile strength was the lowest.

The comparison of the Ot and Cl samples shows that the strength of the Ot sample was lower than that of the Cl sample because of the existence of the coarse-grained layer, which reduced the strength. There was no coarse-grained layer of the Cl sample of the three different plates, and their grain morphology and size were similar; therefore, the difference in strength was only related to the strengthening effect of Al3Zr particles. From the TDSP sample to the DSP sample to the ASP sample, the number density of Al3Zr particles gradually decreased, the average radius gradually increased (shown in Figure 9), and its strengthening effect gradually weakened; therefore, the ultimate tensile strength gradually decreased.

In comparison with 25 °C, the ultimate tensile strength of the three different plate samples at −196 °C is significantly improved. The increase in strength at −196 °C is mainly attributable to the increase in the work hardening capacity. In comparison with 25 °C, the lattice vibration frequency at −196 °C was reduced, and the resistance of moving dislocation was increased; this suppressed the dynamic recovery during the deformation process, resulting in a piling up of abundant activated dislocations and a higher dislocation density. Figure 10 shows the dislocation morphological images of the Cl sample of the TDSP at 25 and −196 °C under 0.12 strain. In comparison with 25 °C, the degree of dislocation piling up of the Cl sample of the TDSP was significantly higher at −196 °C. The geometrically necessary dislocation densities at the same strain of 0.12 were 1.18 × 10^14^ m^−2^ and 1.31 × 10^14^ m^−2^ at 25 and −196 °C, respectively. This explains the increase in tensile strength of the three different plate samples at −196 °C.

### 4.2. Effect of Pretreatment and Cryogenic Temperature on Ductility

According to the Considère criterion, a higher strain-hardening index indicates that the alloy has a higher resistance to local deformation or necking, and higher plasticity [24]. For the Ot samples of the three different plates, the ASP had the largest coarse-grained layer thickness and the smallest strain-hardening index (shown in Figure 5a and Figure 6a); therefore, the ductility was the lowest. The thickness of the coarse-grained layer of the TDSP was the smallest, and the strain-hardening index was the largest (shown in Figure 5a and Figure 6c); thus, the ductility was the highest. The ductility variation law of the Cl samples of the three different plates was the same as that of the Ot sample. From the TDSP to the DSP to the ASP, the ductility gradually increased with the increase in the strain-hardening index.

In comparison with 25 °C, the significantly increased ductility at −196 °C is attributable to the suppression of the dislocation planar slip under the cryogenic temperature. Meanwhile, the grain boundaries and inside grain interiors both participated in the deformation, under cryogenic temperatures, which made the deformation more uniform and coordinated. Figure 11a shows many dense slip bands on the surface of the Cl sample of the TDSP, which indicates that plastic deformation was inhomogeneous at 25 °C, and the deformation degree inside the grain interiors was low; therefore, the sample surface is relatively flat. The number of slip bands on the surface of the sample was significantly reduced and the planar slip was suppressed; meanwhile, the number of grains participating in the deformation increased, and the surface roughness of the sample increased at −196 °C, as shown in Figure 11b. In comparison with 25 °C, the degree of deformation involved inside grain interiors increased significantly at −196 °C, as shown in Figure 12. Figure 12a,b show that the large grid only presents slight deflection after tensile deformation at 25 °C, and the original morphology of the small grid was completely retained; this indicates that the degree of intragranular deformation was low at room temperature. After deformation at −196 °C, small grids were bent as shown by the red closed dotted line in Figure 12d, which indicates that deformation occurred inside grain interiors. Meanwhile, there were few large local misorientation regions at 25 °C, most of which were located at the grain boundary, and there were more small local misorientation regions (shown in Figure 13a), while at −196 °C the large local misorientation regions significantly increased, located in the grain boundary and interiors (shown in Figure 13b). The average value of KAM at 25 and −196 °C were 11.4 and 12.3, respectively. In summary, the planar slip at cryogenic temperatures was suppressed, and the deformation occurred at the grain boundaries and inside grain interiors; this made the plastic deformation more uniform and coordinated, and improved the ductility.

### 4.3. Effect of Pretreatment on the Precipitation of Al3Zr Particles

Precipitation and growth of the precipitates preferentially develop in the dislocation or near the second-phase particle, where the nucleation barrier can be lowered. At the initial stage of homogenization, Zr atoms diffuse to dislocations to moderate their large atomic misfit with the Al matrix [25]. When the Zr atom supersaturation in the dislocation region reaches the precipitation condition, the Al3Zr dispersoids begin to precipitate, and at the dislocation, substrates will try and free themselves from the dispersoids by climbing under thermal activation. The dislocation drags Zr atoms from the solid solution during climbing and Al3Zr forms through fast pipe diffusion. The precipitation of Al3Zr particles through interaction between the Zr solute and dislocation climbing is described as “repeated precipitation on dislocations” [26]. This pattern, in which the precipitates are repeatedly precipitated on the dislocation, requires minimal solute supersaturation.

As the AS sample was annealed before homogenization, the density of dislocations was greatly reduced. Therefore, there were some dislocation-free regions in the intragranular area and near the grain boundary. In the process of homogenization, when the concentration of Zr atoms in this region was insufficient, no Al3Zr particles were precipitated to form a local PFZ, as shown in Figure 8a. In comparison with the AS sample, and owing to a certain number of dislocations in the matrix for the DS sample, the number of Al3Zr particles by repeated precipitation on the dislocation increased during the homogenization, the number of particles in the intragranular area and near the grain boundary increased, and the local PFZ decreased, as shown in Figure 8b. In comparison with that of the DS sample, the TDS sample showed a greater deformation degree, and the number of dislocations in the matrix was further increased; this promoted the precipitation of Al3Zr particles in the intragranular area and near the grain boundary, and eliminated the local PFZ, as shown in Figure 8c. The Al3Zr particle number density was the highest, as shown in Figure 9d. The increased number density of the precipitated phase led to a decrease in particle distances, which shortened the diffusion distance of the solute and accelerated the solute consumption, reducing the saturation of the supersaturated solid solution quicker and shortening the average radius of the dispersoids, as shown in Figure 9. Furthermore, the low concentration of Zr atoms near the grain boundary led to a lower number density of Al3Zr particles than that in the intragranular area. However, the hot deformation before homogenization was able to improve the precipitation, distribution, and uniformity of Al3Zr particles near the grain boundary, in addition to increasing the number density (shown in Figure 9). The TEM image of Al3Zr particles by repeated precipitation on dislocation is shown in Figure 14.

### 4.4. Effect of Pretreatment on Grain Structure

The coherent Al3Zr particles are recognized as dispersoids that can effectively inhibit recrystallization [27,28]. The effectiveness of retarded recrystallization can be measured by the Zener pinning formula [29,30] as follows:(1)$$P_{z}=\frac{f_{v}3\gamma_{GB}}{2r}$$

In the formula, *γGB* is the interface energy, *fv* is the volume fraction, and *r* is the average radius of the dispersoids. This formula indicates that, for a particular dispersoid, the higher the ratio *fv*/*r*, the stronger the resistance to recrystallization. Generally, recrystallized grains are preferentially formed at grain boundaries and second-phase particles, while fine dispersed Al3Zr particles at the grain boundary can effectively hinder the migration of grain boundaries to inhibit recrystallization. In comparison with the AS and DS samples, Al3Zr particles had the largest number density and the smallest size in TDS samples (shown in Figure 8). Furthermore, the distribution of Al3Zr particles was improved and the local PFZ was eliminated, while the distribution was uniform in the intragranular area and near the grain boundary (shown in Figure 7c). Therefore, the thickness of the coarse-grained layer on the surface of the TDSP sample was the lowest after solution treatment, and the fiber grains were better retained (shown in Figure 6c). In comparison with the TDS sample, the number of Al3Zr particles in the intragranular area and near the grain boundary of the DS sample was reduced, along with the presence of local PFZ (shown in Figure 7b), which reduces the resistance to recrystallization. Therefore, the thickness of the surface coarse-grained layer of the DSP sample was increased. For the AS samples, there was a large amount of local PFZ in the intragranular and near the grain boundary. The Al3Zr particles in the AS sample also had the lowest number density, largest size, and highest non-uniformity (shown in Figure 7a). In comparison with those of the DS and TDS samples, the recrystallization resistance was further reduced. Therefore, the thickness of the coarse-grained layer on the surface of the ASP sample was largest.

## 5. Conclusions

In summary, pretreatment before homogenization and an applied cryogenic temperature were investigated on the mechanical properties and microstructure of Al-Cu-Li alloy. The following conclusions can be drawn:(1)The hot deformation pretreatment before homogenization increased the number of dislocations inside grain interiors, promoted the precipitation of Al3Zr particles, improved particle distribution, inhibited local PFZ, and increased recrystallization resistance. In comparison with the AS and DS samples, the Al3Zr particles in the TDS sample had higher number densities, more uniform distributions, and smaller sizes. Therefore, the thickness of the coarse-grained layer of the TDSP was the smallest, and the deformed grains were more retained.(2)Both pretreatment before homogenization and cryogenic temperatures were able to enhance strength. The former promoted the precipitation of Al3Zr particles, enhanced the strengthening effect of the second-phase particles, and reduced the thickness of the coarse-grained layer to increase the strength of the alloy. The rank of the strength can be described as TDSP sample > DSP sample > ASP sample. The latter suppressed the dynamic recovery during the deformation process, which resulted in increasing the dislocation density, enhancing the work hardening effect, and improving the strength of the alloy.(3)Both pretreatment before homogenization and applied cryogenic temperature improved ductility. The former improved ductility as a result of the pretreatment reducing the thickness of the coarse-grained layer and the deformation incompatibility between the coarse and fine grains while increasing the strain-hardening index. The rank of the elongation can be described as TDSP sample > DSP sample > ASP sample. The latter improved ductility due to the suppression of planar slip and the strengthening of grain boundaries at a cryogenic temperature, which promoted the participation of intragranular deformation and made the deformation more uniform.

## Figures and Tables

**Figure 1 materials-14-04873-f001:**
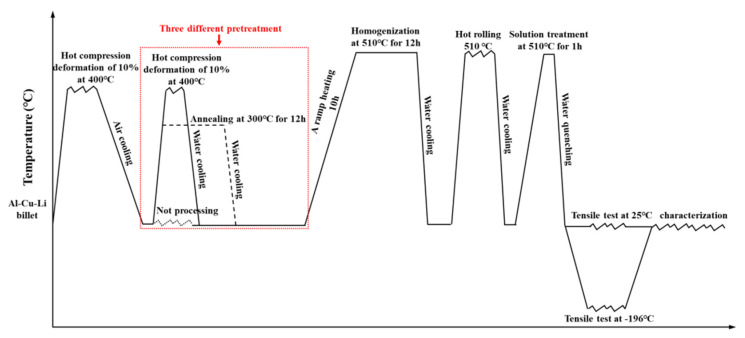
Schematic diagram for the whole heat treatment and test conditions.

**Figure 2 materials-14-04873-f002:**
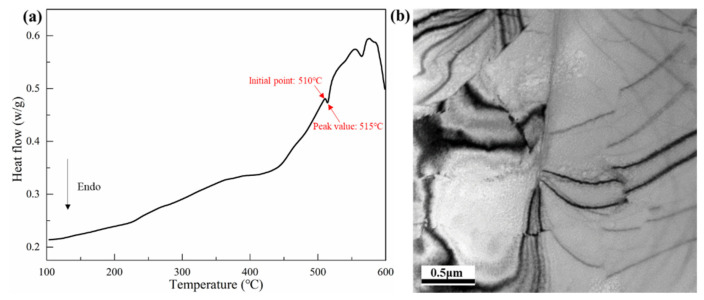
(**a**) DSC scan of as-cast samples of Al-Cu-Li alloy; (**b**) TEM images of the as-cast Al-Cu-Li alloy.

**Figure 3 materials-14-04873-f003:**
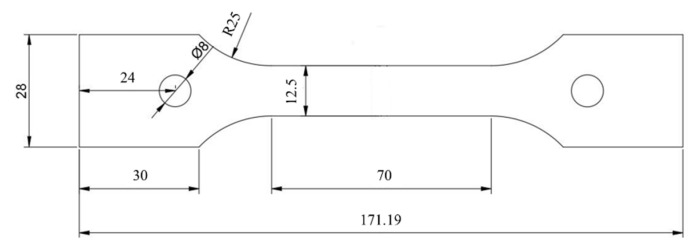
Dimensions of the uniaxial tensile-test sample (mm).

**Figure 4 materials-14-04873-f004:**
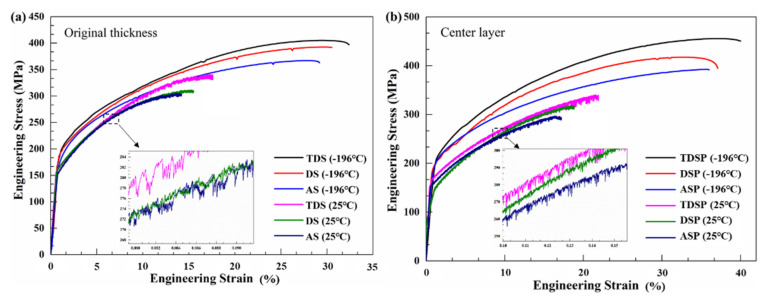
Engineering stress versus strain curves of the Ot and Cl samples of three different plates at different temperatures: (**a**) 25 °C; (**b**) −196 °C.

**Figure 5 materials-14-04873-f005:**
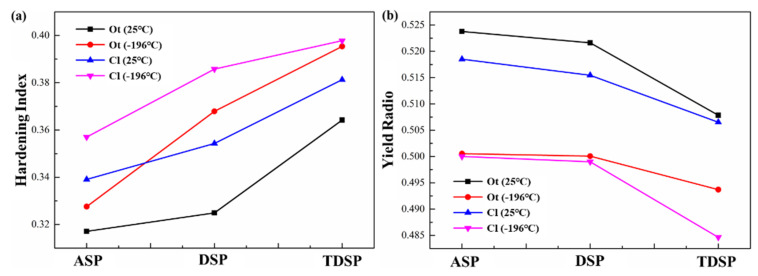
The n values and yield-strength ratios of the Ot and Cl samples of three different plates at different temperatures: (**a**) 25 °C; (**b**) −196 °C.

**Figure 6 materials-14-04873-f006:**
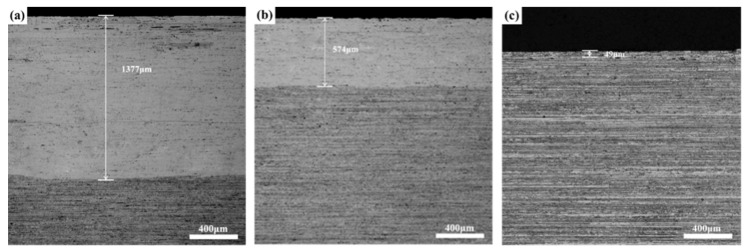
OM images of three different plate samples: (**a**) ASP sample; (**b**) DSP sample; (**c**) TDSP sample.

**Figure 7 materials-14-04873-f007:**
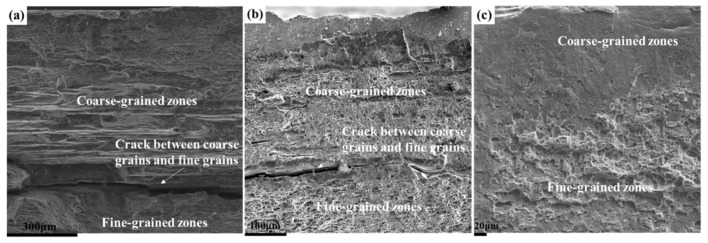
Tensile fracture images of Ot specimens of three different plates near the surface: (**a**) ASP; (**b**) DSP; (**c**) TDSP.

**Figure 8 materials-14-04873-f008:**
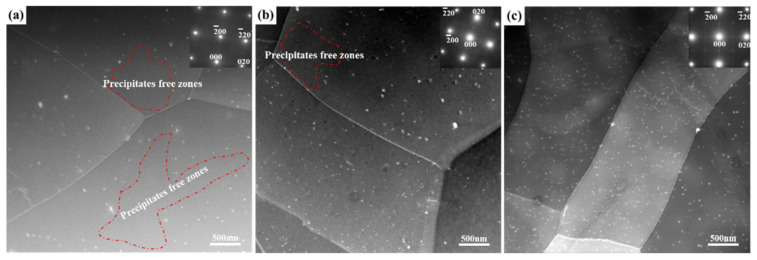
STEM images of Al3Zr distributions in the intragranular and grain boundary of three different homogenized samples: (**a**) AS sample; (**b**) DS sample; (**c**) TDS sample.

**Figure 9 materials-14-04873-f009:**
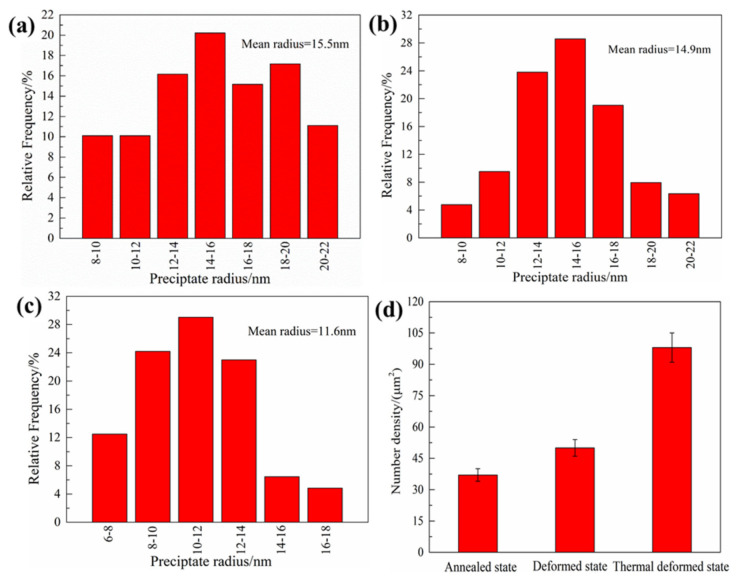
Histogram of Al3Zr dispersoids size distribution and number density of three different homogenized samples: (**a**) AS sample; (**b**) DS sample; (**c**) TDS sample; (**d**) number density.

**Figure 10 materials-14-04873-f010:**
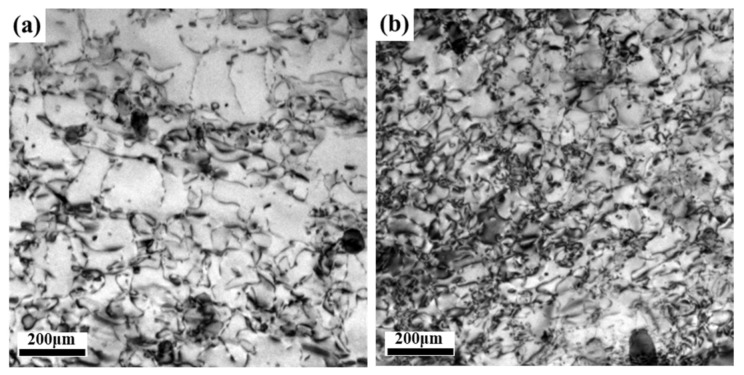
Dislocation morphology images of the Cl specimen of TDSP tensioned to a fixed strain of 0.12 at (**a**) 25 °C and (**b**) −196 °C.

**Figure 11 materials-14-04873-f011:**
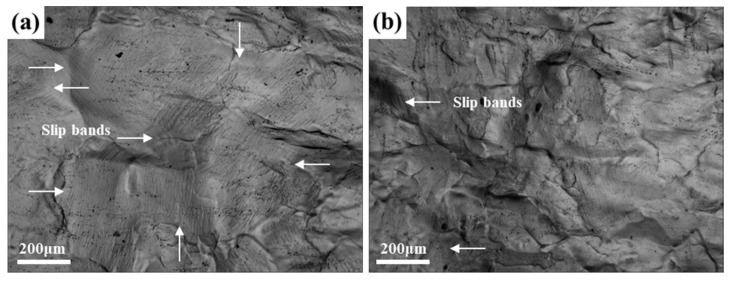
OM observations on the surface of the Cl sample of TDSP tensioned to a strain of 0.12 at different temperatures: (**a**) 25 °C; (**b**) −196 °C.

**Figure 12 materials-14-04873-f012:**
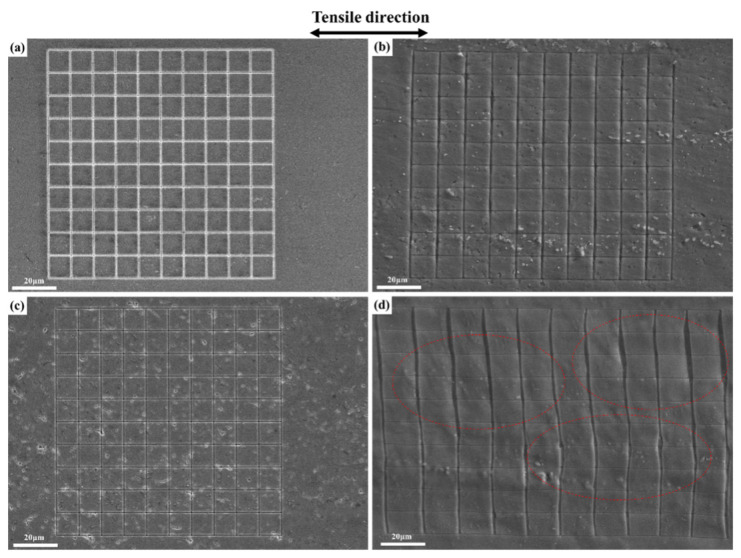
SEM observations on the surface of the Cl sample of TDSP before and after deformation to the strain of 0.12 at different temperatures: (**a**) before deformation at 25 °C; (**b**) after deformation at 25 °C; (**c**) before deformation at −196 °C; (**d**) after deformation at −196 °C.

**Figure 13 materials-14-04873-f013:**
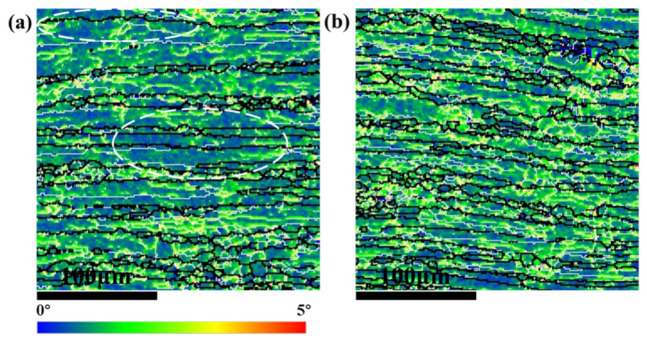
KAM maps showing the local misorientation gradients on the surface of the Cl sample of TDSP tensioned to 0.12 strain at (**a**) 25 °C and (**b**) −196 °C.

**Figure 14 materials-14-04873-f014:**
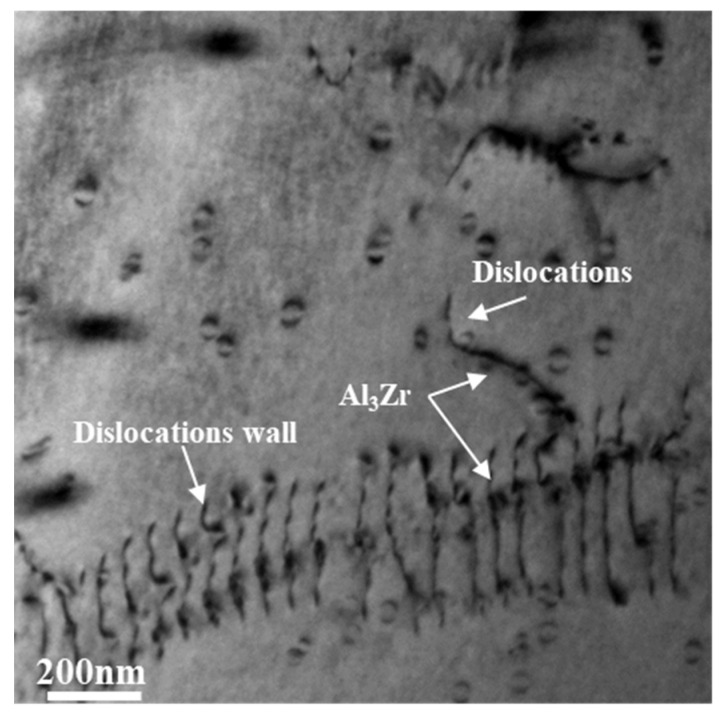
Al3Zr particles by repeated precipitation on dislocation.

**Table 1 materials-14-04873-t001:** Chemical composition of Al-Cu-Li alloy (mass fraction/%).

Si	Fe	Cu	Mn	Mg	Zr	Ag	Li	Al
0.09	0.13	3.72	0.23	0.44	0.12	0.31	1.06	Bal

**Table 2 materials-14-04873-t002:** Tensile properties test results of the Ot and Cl samples of the three different plates at different temperatures.

Sample	Different Positions	Temperature	Ultimate Tensile Strength	Elongations
		°C	MPa	%
TDSP	Ot	25	340	17.6
		−196	405	32.4
	C1	25	343	21.9
		−196	455	39.9
DSP	Ot	25	311	15.5
		−196	392	30.8
	C1	25	319	18.8
		−196	417	37.1
ASP	Ot	25	296	14.2
		−196	367	29.2
	Cl	25	305	17.2
		−196	392	36.0

## Data Availability

Not applicable.

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
