# Peer review of "Effect of Pretreatment and Cryogenic Temperatures on Mechanical Properties and Microstructure of Al-Cu-Li Alloy"

_materials, 2021, doi:10.3390/ma14174873_

Round 1

Reviewer 1 Report

The article discusses three different pretreatments carried out before homogenization and their influence on the mechanical properties and microstructure of the cast Al-Cu-Li alloy. The composition of the article meets the journal's recommendations and includes the following sections: Abstract, Introduction, Materials and Methods, Results (divided into three subsections), Discussion (divided into four subsections), Conclusions, Acknowledgments, and References. This is a worthwhile and very well-prepared manuscript in the reviewer’s opinion.

Some minor improvements are recommended:

  • The legend on the Figure 4 should be verified,
  • Figure 7 and 11 are completely illegible, a font color change is recommended,
  • Line 317 and 353: subsection 4.3 and 4.4 titles have the same description.

Author Response

Dear reviewer:

Thank you very much for your letter and  comments concerning our manuscript entitled “Effect of pretreatment and cryogenic temperatures on mechanical properties and microstructure of Al-Cu-Li alloy” (ID materials-1310201). Those comments are all valuable and very helpful for revising and improving our paper, as well as the important guiding significance to our researches. We have studied the comments carefully and made corrections which we hope meet with approval. The main corrections in the paper and the responds to the reviewer’s comments are as below.

(1) Response to comment 1.

Response: According to the suggestion, the legend on the Figure 4 have been revised.

(2) Response to comment 2.

Response: According to the suggestion, a font color of Figure 7 and 11 have been revised.

(3) Response to comment 3.

Response: According to the suggestion, the subtitle 4.4 have been replaced by “Effect of pretreatment on grain structure”.

Sincerely yours

Jin Zhang

Reviewer 2 Report

This paper explains the effect of pretreatment and cryogenic temperatures on the mechanical properties and microstructure of Al-Cu-Li alloy. The results are all interesting and novel, but the experimental data to support the conclusion is not clear and insufficient. More profound descriptions are needed. 

  1. The authors compared three pretreatment groups (annealed, deformed, thermal deformed state) with original thickness and centered layer. Why did you compare the original thickness and centered layer? The reasons for this comparison were not stated clearly.

  1. The relationship between strength and ductility is known as trade-off property. With cryogenic temperature (-196 degree C), both the strength and ductility have been increased compared to the 25 degree C. The authors stated the reason for strengthening was due to the increase of precipitate number density and the reason for the increase of the ductility was due to the suppression of the dislocation planar slip. How did you find slip bands with OM observations? It seems more slip bands are present in Fig. 11(b). Furthermore, the explanation of the reason for the ductility increase is not sufficient.

  1. In Al-Cu-Li alloy, what particles are mainly precipitated? The authors mentioned only the Al3Zr, but what about Al2Cu and other precipitates? The effect of nano-sized Al2Cu precipitates should be discussed.

  1. Subtitle 4.4 and 4.3 are the same.

  1. The caption for Fig. 9(d) is missing.

Author Response

Dear reviewer:

Thank you very much for your letter and comments concerning our manuscript entitled “Effect of pretreatment and cryogenic temperatures on mechanical properties and microstructure of Al-Cu-Li alloy” (ID materials-1310201). Those comments are all valuable and very helpful for revising and improving our paper, as well as the important guiding significance to our researches. We have studied the comments carefully and made corrections which we hope meet with approval. The main corrections in the paper and the responds to the reviewer’s comments are as below.

(1) Response to comment 1.

Response: According to the suggestion, the reasons is that the surface layer of the plate is easy to produce coarse grain, and the coarse grain layer is harmful to the mechanical properties. Therefore, in order to fully explore the effects of pre-deformation and cryogenic temperature on the properties and microstructure of Al-Cu-Li alloy, the original thickness and the central layer were compared.

(2) Response to comment 2.

Response: According to the suggestion, we are sorry for the ambiguous expression of find slip bands with OM observations. To observe the surface morphology of the tensile samples after deformation, the sample needs to be grinded and polished before tensile test. Finally, the deformed sample is directly observed by optical metallography.

According to the suggestion, Fig. 11(a) have been replaced.

We are sorry for the ambiguous expression of the reason for the ductility increase. According to the suggestion, the explanation of the reason for the ductility increase have been supplemented.

(3) Response to comment 3.

Response: According to the suggestion, we are sorry for the ambiguous expression of the precipitation of the alloy. In this study, the object of research is W-tempered Al-Cu-Li alloy. The nano-sized Al2Cu and T1 precipitates have been completely dissolved into the matrix in the solid solution stage, while the melting point of Al3Zr particle is higher than 510℃, so only Al3Zr is observed.

(4) Response to comment 4.

Response: According to the suggestion, the subtitle 4.4 have been replaced by “Effect of pretreatment on grain structure”.

(5) Response to comment 5.

Response: According to the suggestion,the caption for Fig. 9(d) have been added.

We appreciate for Editors/Reviewers’ warm work earnestly, and hope that the correction will meet with approval. Once again, thank you very much for your comments and suggestions.

Sincerely yours

Jin Zhang

Round 2

Reviewer 2 Report

The authors addressed the comments properly, therefore, I recommend publication as it is.